# Grid-Friendly Active Demand Strategy on Air Conditioning Class Load

**Jian-hong Zhu *** **, Juping Gu and Min Wu**

Institute of Electrical Engineering, Nantong University, Nantong 226019, China; gu.jp@ntu.edu.cn (J.G.); 1912310002@stmail.ntu.edu.cn (M.W.)

* Correspondence: jh.zhu@ntu.edu.cn; Tel.: +86-135-1520-1945

**Abstract:** The growing number of accessed energy-efficient frequency conversion air conditioners is likely to generate a large number of harmonics on the power grid. The following shortage in the reactive power of peak load may trigger voltage collapse. Hence, this conflicts with people's expectations for a cozy environment. Concerning the problems mentioned above, an active management scheme is put forward to balance the electricity use and the normal operation of air conditioning systems. To be specific, first, schemes to suppress the low voltage ride through (LVRT) and harmonic are designed. Second, to deaden the adverse effects caused by nonlinear group load running on the grid and prevent the unexpected accidents engendered from grid malfunction, the dynamic sensing information obtained by an online monitor is analyzed, which can be seen as an actively supervised mechanism. The combined application of active and passive filtering technology is studied as well. Third, the new energy storage is accessed reliably to cope with peak-cutting or grid breaking emergencies, and the fuzzy control algorithm is researched. Finally, system feasibility is verified by functional modules co-operation simulation, and active management target is achieved under scientific and reasonable state-of-charge (SOC) management strategy.

**Keywords:** air conditioning group load; grid friendly; active demand; storage; coordinated control

## 1. Introduction

As the main facility of a building, air conditioning configurations grow quickly, accompanied by increased buildings, which are used by more and more enterprises and groups because they can provide comfortable surroundings, such as air temperature, relative humidity, and cleanliness. However, the maximum load capacity designed on the central air conditioning is according to the local maximum temperature of the working environment by a national standard, but it works mostly at 70% of the full load, which is massive redundancy in the system design [1]. This means that the demand for building energy consumption is improved continuously.

Consequently, massive harmonics appear with increased inverter air conditioners (IRC) connected to the grid [2]. The high-order harmonics may cause false triggers on other load equipment, affecting the normal operation and even run out of control [3]. On the other hand, frequency conversion may increase the power loss of transmission lines, dwindle transformer power capacitors, lead to overheating of the equipment, and reduce equipment utilization life and economic benefits [4]. Techniques such as power factor correction rectifiers are also used but do not consider total harmonic processing on the grid [5]. The detection and processing technology of grid harmonics must be studied [6]. Relevant harmonic suppression technology is proposed and improved continuously [7].

When the contradiction is very prominent between power supply and demand, the active demand-side management needs to be promoted, which is a set of flexible and efficient active management schemes to make load demand changes stable. Power quality can be improved under the measurement of the power grid and load parameters, while some policy incentives or technologies are provided as well [8]. Measures for using electricity safely and improving the central air conditioning cooling or chilling water system would be imperative for the host to operate efficiently and reduce energy consumption. Some experts and scholars have proposed innovative operation measures for energy-saving technologies [9]. The authors of literature [10] proposed temporary cut out measures of air condition load when receiving an emergency request from the smart grid or to circumvent the impact of the grid by shutting down the number of the cooling group through a power demand optimization scheme, which did not sacrifice too much comfort.

With the development and utilization of new energy, coordinated scheduling of related equipment, the power mismatch between supply and demand of air conditioning load is solved to the utmost extent [11], where new energy is used to achieve the goal, but the impact of air conditioning on the grid is not considered. Of course, to achieve better goals of energy-saving and comfort and minimize the impact of nonlinear load operation on grid quality, LVRT and harmonic control technology are essential. Some experts have proposed some related techniques based on the cost function; active consumption behavior is stimulated to deal with peak load shifts and reduce the unstable operation of the grid [12]. From the perspective of operating characteristics of the air conditioning group load, designing a grid-friendly system composed of harmonic suppression and the LVRT mechanism at the load side is necessary [13]. In addition, the lack of reactive power at peak load can cause voltage fall and even lead to the entire system collapse, which undoubtedly has a great impact on the normal daily life of the people. Therefore, reactive power compensation technology also plays a pivotal role in the safe and stable operation of the power grid [14].

Hence, the goal of active demand-side management is proposed to cater to the energy crisis and alleviate power supply demand, mainly by changing the way of electricity consumption and improving the benefit of terminal electricity, making full use of the functions and operation characteristics of the energy storage system. On the one hand, measures are taken to reduce the electricity demand during the peak load period of the grid, as well as increase the power demand during the valley period, to decrease the demand for grid reserve capacity. On the other hand, active demand-side management is carried out actively to suppress harmonics of the grid, protect electrical equipment during grid malfunction or LVRT, and ensure the normal operation of the load at the same time. By designing reactive power compensation and providing that on-grid restoration, grid operation is prevented from collapse and system power loss, maximizing social benefits. In the following, research work is developed around active harmonic suppression, LVRT, and reactive power compensation. The relevant schemes are designed, and the feasibility is verified by simulation.

## 2. Active Demand Strategy Design

Based on experience, the factors affecting human comfort include atmospheric temperature, humidity, air pressure, illumination, wind, clothing, and individual differences. Since not all indicators have equal impacts on human comfort, an improved non-dimensional regression model is proposed by coefficient parameters method, where several main factors are selected, and coefficient and modified parameters are designed according to their differences in the impact on human comfort [15,16]. Among the meteorological factors that affect the comfort level of the human body, the temperature is in the first place, next is humidity, followed by wind direction and wind speed. To make air-conditioned meet the requirements of human comfort, the water system, wind system, and cold and heat source must be coordinated and properly controlled. Generally, the following objectives such as air quality (freshness), comfort (temperature, humidity, and ventilation), and energy-saving (minimum power consumption) must be taken into account in air conditioning control. Thus, the multi-objective working model should be expressed as Formula (1).

$$\begin{cases} ssd = (1.818t + 18.18)(0.88 + 0.002f) + (t - 32)/(45 - t) - 3.2\sqrt{v} + 18.2 \\ \text{OPT(RAF)} \\ \text{Min}(\sum W_i) \end{cases} \quad (1)$$

Among them, $ssd$ is the human body comfort index; the formula definition of comfort level is not the same according to a different classification. $t$ is the average temperature, $f$ is the relative humidity, and $v$ is the wind speed. The comfort formula is an empirical formula based on long-term statistics of the project. $\sum W_i$ is the sum of energy consumed by motor drag systems. $W_i$ is the energy consumed by the pump and the fan. RAF is an abbreviation of relative air freshness; the best comfort index is 59–70 and the human body feels the most comfortable and acceptable. In the selecting process of many parameters, the average temperature $t$ and relative humidity $f$ can be freely selected according to the actual situation, and the wind speed $v$ is determined according to the control demand. From the perspective of meteorology, the human body feels more comfortable when the temperature is between 18 and 20 Celsius and the relative humidity is between 50% and 60%. At present, the human body comfort index adopted in Jiangsu Province has nine grades from −4 to 4, of which level 0 is the most comfortable and most acceptable [17]. It can also be seen from the formula that the air conditioning operation is mainly the coordination work of the fans, pumps, compressors, and heaters. Therefore, in the same external environment, the higher is the human comfort control index, the higher is the energy consumption of air conditioning equipment. The air conditioning group load simulation with typical nonlinear characteristics motors group as shown in Figure 1.

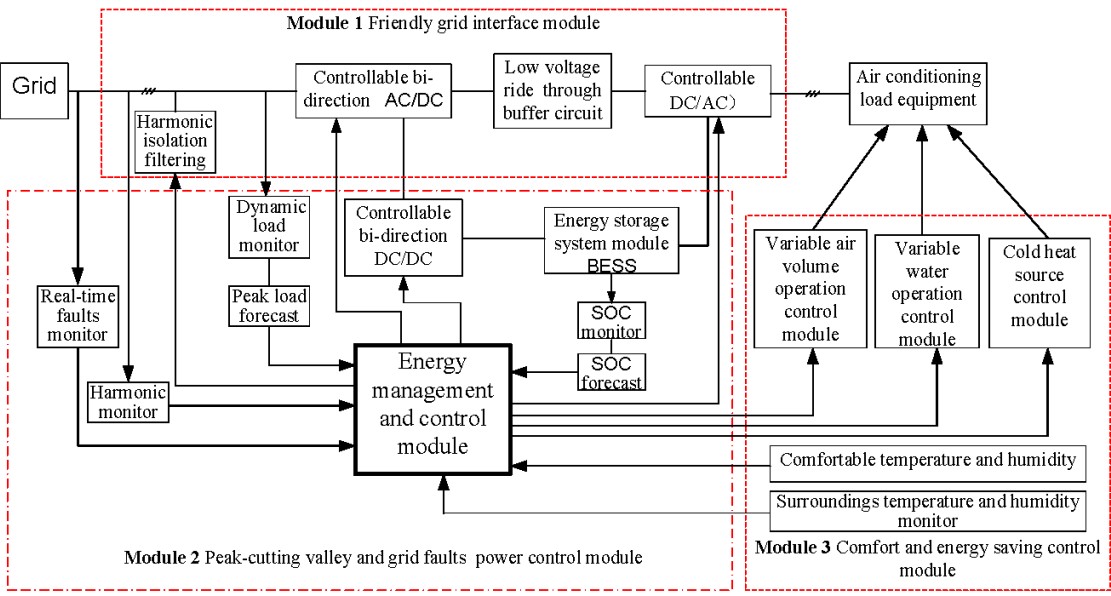

**Figure 1.** The module structure of active demand management.

The left side of the figure shows the power quality monitor of the current harmonics and voltage distortion caused by the group load operation of the air conditioning system. The middle is the inverter system that supplies power to the air conditioning system. The right part represents the load of the air conditioning group. It includes a variable water volume system, a variable air volume system, and a cold–heat source system. They are respectively composed of different motor machines, yet the sensors and data analysis unit in the figure are designed to realize active energy monitoring. Based on the analysis of load operation characteristics, a friendly scheduling design on the energy-saving operation is carried out. The hardware functional modules included in the overall active demand strategy are made of harmonic suppression and governance, LVRT control and reactive power compensation, peak shaving, and emergency control.

The Z-type filter and harmonic prevention circuit are designed on the DC bus side between the power supply and load. It buffers current shock whenever the nonlinear load group starts or sudden changes from the power grid. A grid monitor is designed so that hardware chain action can work at the same time as the severe emergency grid faults such as LVRT. The energy storage and grid power supply are switched to each other by the interlock linkage circuit. Besides, combined with load forecasting, battery energy storage, and load peak-to-peak management, a bidirectional charge–discharge circuit is designed between the DC bus and the energy storage battery. The energy is made full use during the LVRT period, and the stored energy is released during peak hours to achieve partial peak load shift, ensuring the normal operation of air conditioning. Once the energy of the battery is insufficient to maintain the load in normal operation at the peak load period, and indoor temperature and humidity are within a certain range of the human body fitness, suspension strategy is performed due to delay characteristics of the central air conditioning system. The active management technology of the group load is provided by the means of flexible huff–puff characteristics of the energy storage system. On the one hand, the normal power supply under the grid fault condition is guaranteed. On the other hand, the shifting peak and valley function is realized through energy scheduling technology. Of course, key technologies such as energy storage management and converter control technology must be given as far as possible. In the following, the topic is discussed around the active demand-side management technology from the aspects of harmonic processing, LVRT, and reactive power compensation.

## 3. Harmonic Suppression Technology

### 3.1. Air Conditioning Group Load Operation Characteristics

The air-conditioning group is nonlinear, so the rectifier circuit is composed of power electronic devices and the multi-motor combination is selected in the system to simulate the operation model. During the peak period of air conditioning working, multiple air conditioners start running at the same time. The load characteristics are shown as the top waveform in Figure 2b; load operation would bring harmonics to the grid, and the current distortion rate generated is very large, up to 14.12%. According to the standards proposed in IEEE Std 929-2000 and IEEE Std. P1547, the control parameters of each odd harmonics and Total Harmonic Distortion (THD) rate in the power system are shown in Table 1. The non-sinusoidal current generated from the air conditioning nonlinear group load is the harmonic source on the utility grid. The voltage drop is generated on the circuit impedance, which is superimposed on the sine wave voltage, resulting in waveform distortion of grid voltage that affects the power quality [18]. Therefore, it is important to monitor and improve the electrical energy characteristics of the air conditioning nonlinear load operation. As shown in Table 1, the 5th, 7th, and 11th harmonic current components are larger, and the 5th and 7th harmonics account for 10.22% and 5.34%, respectively, which are less than 4%, meeting the requirements of $n < 11$, while harmonic current components of 11, 13 and 17 sum up to 19%, more than 2%, which does not meet the requirements. The total harmonic distortion rate is far beyond the requirements. Since the system only selects several motor models to simulate the nonlinear group load, there is still a large gap compared with the actual large-scale load. If the harmonics are not processed, the consequences are serious. Therefore, it is necessary to monitor the quality of the incoming grid current by THD, and it is necessary to take measures to suppress harmonics and provide high power quality. In Figure 2, FFT harmonic analysis results show that, after APF (active-passive filter) or PF (passive filter) processing, harmonics are suppressed to a certain extent. In the following, the operation principle of PF and APF are analyzed firstly.

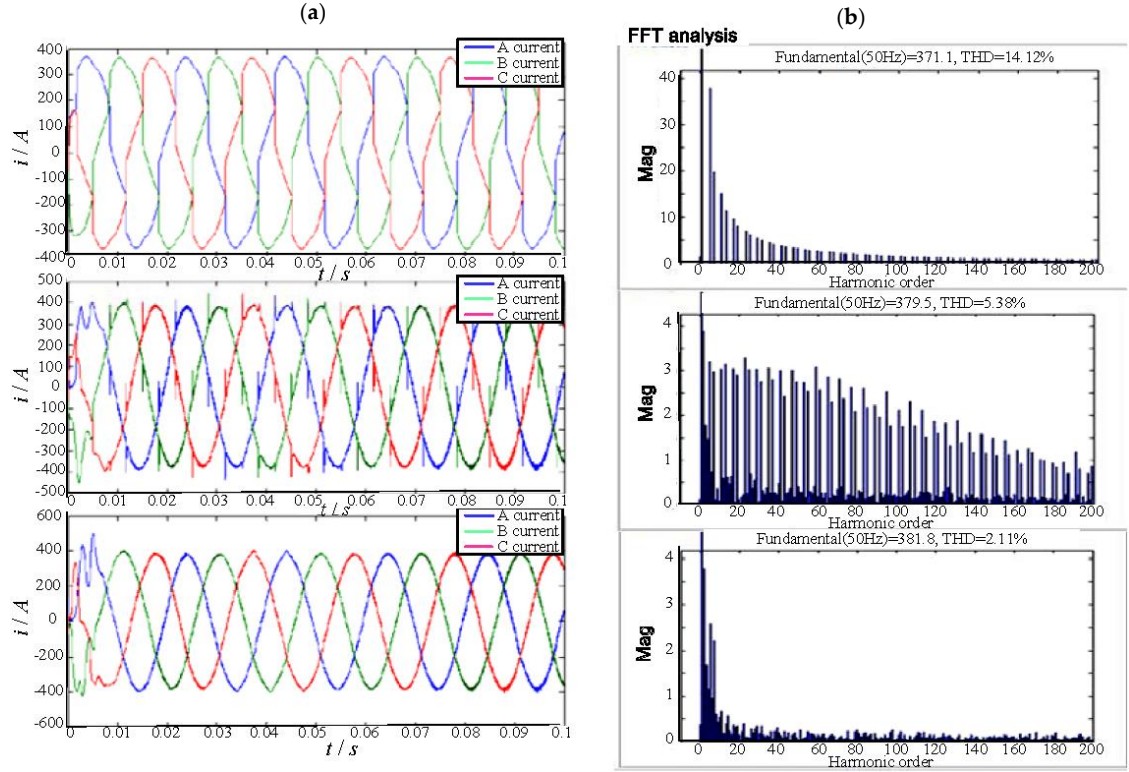

**Figure 2.** (**a**) Current before and after filtering (left); (**b**) FFT harmonic before and after filtering (right).

**Table 1.** Harmonic constraint standard and actual harmonic of grid.

| Odd Order | $n < 11$ | $11 \le n \le 17$ | $17 < n < 23$ | $23 < n < 35$ | $n > 35$ | THD |
|---|---|---|---|---|---|---|
| Maximum harmonic constraint from standard | 4.0% | 2.0% | 1.5% | 0.6% | 0.3% | <5% |
| Actual harmonics | 2.2% | 19% | 23% | 9.5% | 0.6% | 14.12% |

### 3.2. Filtering Technology Combining Active and Passive Filtering

Harmonic detection is the primary link to realize APF control. The accuracy and response speed of harmonic signal detection directly affect the compensation effect. If the detected harmonic component is not accurate, the subsequent control will be meaningless. Thus, instantaneous reactive power harmonic current detection mode and current hysteresis control mode are used [19]. Improved APF is used to filter out most of the harmonics, while PF is used to filter out other small portions.

#### 3.2.1. APF Technology

The APF is generally a DC voltage source of the inverter. The power supply here is the energy storage system. Unlike the traditional passive compensation method, the APF injects harmonics actively. The main circuit is a controlled inverter composed of power electronic devices. By collecting the load device current, the harmonic currents are extracted and used to control the switch tubes.

The reverse polarity harmonic current signals generated by the inverter are injected into the grid, together with the load current to eliminate unwanted components. The APF includes a main circuit module, a nonlinear load module, a harmonic detection module, and a compensation current control module. The nonlinear load side is fixed up three-phase current–voltage measurement module, measurement results are input to the harmonic detection circuit module to obtain command current. The compensation current control module is used to compare and control the main circuit switching device, to compensate for the system harmonics.

Harmonic Current Detection

The harmonic current detection method is based on instantaneous reactive power [20], as shown in Figure 3. Three-phase current in the grid are $i_a, i_b, i_c$, fundamental current are $i_{af}, i_{bf}, i_{cf}$, harmonic current are $i_{ac}^*, i_{bc}^*, i_{cc}^*$, power voltage is $e_a$. $C_{pq}$ is a power–current variation matrix, LPF is low-pass filtering, the three-phase grid voltage is converted by C3-2, and is filtered by LPF to obtain a harmonic signal followed by comparing with the fundamental wave.

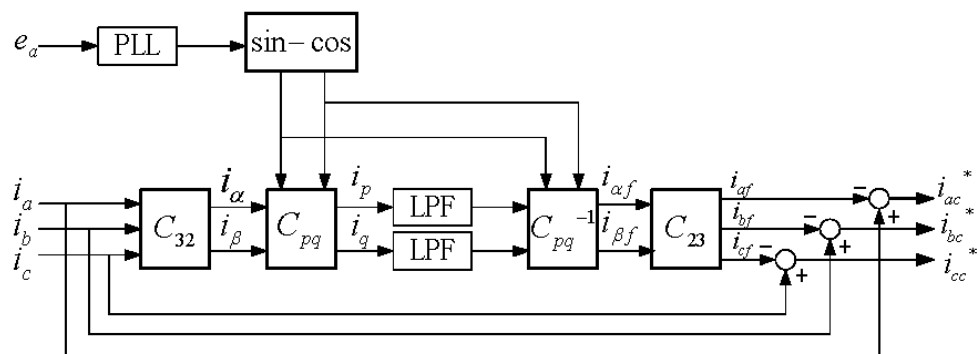

**Figure 3.** $i_p$–$i_q$ based harmonic detection method schematic.

Current Hysteresis Tracking Control

The traditional hysteresis control principle utilizes the difference between the compensation current and the command current through the loop width, that is, constructs a hysteresis loop, 0 as the center, H, and -H as the upper and lower limits to ensure the difference within the hysteresis range. Then, the switching action of the converter is achieved. The improved technique uses a voltage-space vector double hysteresis current tracking control method to reduce both switching frequency and current tracking error at the same time [21]. Higher harmonic components can be reduced while ensuring the current response speed. The voltage SVPWM (Space Vector Pulse Width Modulation) synthesizes the reference voltage vector accurately, which improves the DC voltage utilization rate while maintaining a constant switching frequency. The idea of double hysteresis current control based on the space voltage vector is shown as Formula (2).

$$U_k = RI_C + L\frac{dI_C}{dt} + e \tag{2}$$

Among them, $U_k$ is the inverter output voltage vector, $I_C$ is the corresponding output current vector, and $e$ is the terminal voltage. Error current vector $\Delta I$ is defined as the difference between the current reference vector $I_C^*$ and actual output current vector $I_C$, as Formula (3).

$$\Delta I = I_C^* - I_C \tag{3}$$

Substituting Formula (3) into Formula (2), and ignoring the effect of equivalent resistance, results in Formula (4).

$$L\frac{d\Delta I}{dt} = \left(L\frac{dI_C^*}{dt} + e\right) - U_k \tag{4}$$

Define reference voltage vector $U^*$, as in Formula (5).

$$U^* = L\frac{dI_C^*}{dt} + e \tag{5}$$

which can be simplified as Formula (6).

$$L\frac{d\Delta I}{dt} = U^* - U_k \tag{6}$$

When the current error vector is in a range other than the outer loop, the current error is large. Then, fast control is required, and only the partition of the error current vector is considered. To rapidly reduce the current error, the central axis of the range where the error current vector is located should be selected. At this time, regardless of the reference voltage vector $U^*$, $U^* - U_k$ will be a component that points to the origin of the coordinate along the central axis of the partition $\Delta I$, and this component is the largest of all choices. The improved technique simulation diagram is shown in Figure 4.

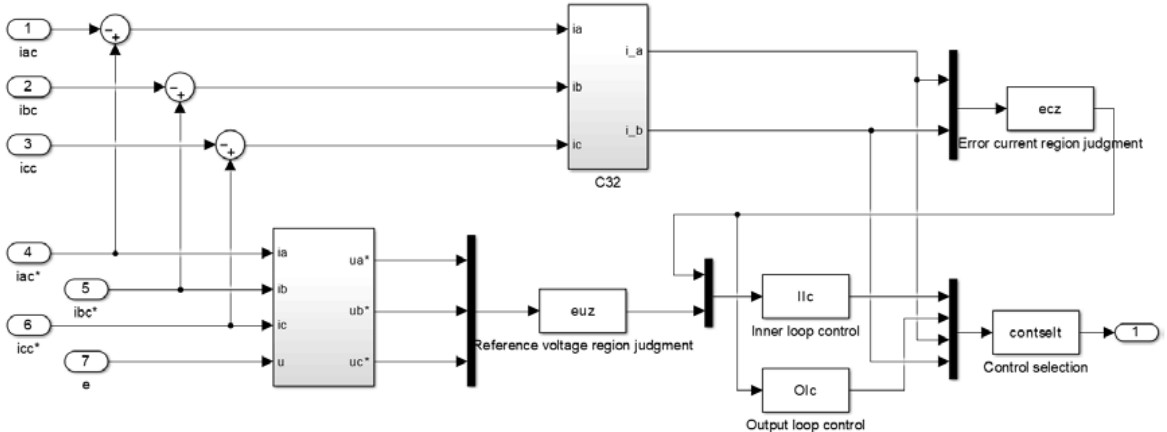

**Figure 4.** Improved hysteresis control simulation model.

Waveforms before and after APF harmonic processing are shown in the middle of Figure 2a and the middle of Figure 2b. It can be seen in the figure that the distortion rate of the current has dropped significantly after APF, and is near 5.38%. Among them, the 5th, 7th, and 11th harmonics are obviously reduced, and the harmonics can basically be controlled. However, the THD still does not meet the standard of 5% or less; there are still large spikes and burrs in the filtering cycle, and the waveform is not very perfect.

### 3.2.2. PF Technology

PF is composed of components such as capacitors, inductors, and resistors. System PF adopts the LCL structure. The remaining frequency distribution not filtered out by APF is analyzed, RLC parameters are calculated to obtain suitable values. The cutoff frequency of the LC filter is determined first. It must be much smaller than the lowest harmonic frequency contained in the voltage, and at the same time much larger than the fundamental frequency [22]. The cutoff frequency is selected as Formula (7).

$$10f_1 < f_L < f_{har(min)} \tag{7}$$

$f_1$ is the fundamental frequency and $f_{har(min)}$ is the lowest harmonic frequency. The cutoff frequency of the simulation is determined as 2.25 kHz. According to the calculation, the filter capacitor is $2.82 \times 10^{-4}$ C and the filter inductance is $1.776 \times 10^{-5}$ H. The waveforms after APF and PF are shown in Figure 2a (bottom) and harmonics analysis Figure 2b (bottom). The FFT analysis chart and Table 2 show that the total distortion rate has dropped to 2.11%, which is less than 4%.

**Table 2.** THD of odd harmonic before and after filter.

| Harmonic/Number | 5 | 7 | 11 | 13 | 15 | 17 | 19 |
|---|---|---|---|---|---|---|---|
| **Before filtering** $i_a/A$ | 37.94 | 19.81 | 15.12 | 11.48 | 9.50 | 8.07 | 37.94 |
| **APF filtering** $i_a/A$ | 3.20 | 2.97 | 3.03 | 3.15 | 3.04 | 2.91 | 3.20 |
| **Active and passive filtering** $i_a/A$ | 2.58 | 2.21 | 0.68 | 0.41 | 0.38 | 0.35 | 2.58 |

It can be also seen that APF first eliminates a large number of harmonics, and PF eliminates a small number of specific components harmonics. In addition to designing of the low-voltage snubber circuit of the inductor–capacitor structure, an energy storage system as an emergency on-grid malfunction is also designed.

## 4. Active Prevention Strategy on LVRT and Reactive Compensation

Based on the approximate circuit model design of the nonlinear group load of air conditioning, the power characteristics of the load under the ideal grid condition are monitored. However, under actual conditions, the power grid is often affected by external disturbances, the short-circuit fault has the greatest influences, and the voltage will fall, which will adversely affect the electrical equipment, and even cause damage to the equipment. In general, the common method is to cut off the electrical equipment from the grid to protect the local equipment. However, with the proportion of nonlinear loads increasing [23], the traditional method will cause the grid voltage and frequency to fluctuate widely, resulting in the collapse of the whole grid system. Therefore, the energy storage system is more suitable for the power equipment to own LVRT capability.

### 4.1. Energy Storage System Based on Isolated DC/DC Converter

4.1.1. The Topology of the Energy Storage System

To enable the system LVRT ability, the internal structure of the IRC can be skillfully utilized, and the energy storage system is added to the DC bus of the converter. The DC side and the energy storage system are joined together by a bidirectional DC converter, which is an isolated structure to achieve LVRT control, as shown in Figure 5.

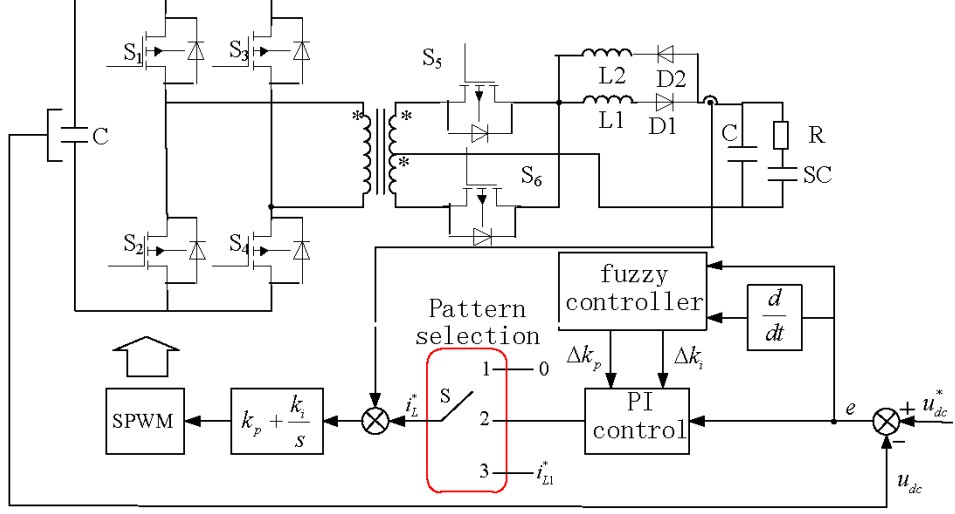

**Figure 5.** Fuzzy adaptive PI control of energy storage.

The DC side of the full-bridge can withstand the change of high power, and the energy storage system of the super-capacitor is connected to the push–pull side. The equivalent model of the super-capacitor uses a series connection of capacitor SC and internal resistance R to function as a bidirectional flow of energy. During the process of frequent charge and discharge of the energy storage system, transient fluctuations are easily induced by the sudden change of the inductor current. Therefore, a freewheeling inductor L2 and a diode are connected in parallel on the push–pull side to make a direction change of the current easy of the conduction. When the bidirectional DC converter operates in the buck mode, the current flows from the loop of L1. In the Boost mode, the current flows from the loop of L2, which can effectively solve the shortcoming of transient fluctuation of the current during the charge and discharge switching of the single inductor. In this paper, the isolated bidirectional DC/DC converter is used to realize the control of LVRT. When the DC bus voltage is greater than the rated voltage during the grid malfunction, the bidirectional DC converter works in the buck mode. At this time, the switch tubes S5 and S6 are equivalent. The diode plays the role of synchronous rectification. By controlling the conduction and deactivation of S1–S4, the excess energy on the DC side is stored in the right energy storage element. During the complementary conduction of the switching tubes S1 and S4, as well as S2 and S3, the DC-side voltage is stored in the inductor L1 through the single-phase full-bridge and the transformer, and the inductor L1 passes through S5 and S6 during the simultaneous shutdown of S1–S4. Perform freewheeling to release energy. When the DC bus voltage is less than the rated voltage, the converter operates in the boost mode. At this time, the switch tubes S1–S4 correspond to the diodes, which play the role of synchronous rectification. The purpose of the energy storage system to transfer energy from the push–pull circuit to the DC side is achieved by controlling the switching tubes S5 and S6. During the alternate conduction of the switch tubes S5 and S6, the energy storage element transfers energy to the left DC bus through the push–pull converter and the transformer, where the inductor L2 stores energy and the current rises. During the simultaneous conduction of S5 and S6, the inductor L2 continues to flow through S5 and S6, releasing energy and decreasing the current. Therefore, the push–pull full-bridge bidirectional converter can operate in a step-up or step-down mode according to changes in the DC-side voltage during a grid malfunction, maintaining the DC bus voltage in the allowable range, and achieving LVRT of the entire system.

### 4.1.2. Parameter Design of the Energy Storage System

Super-Capacitor Capacity Design

As the most important part of the energy storage system, the capacity configuration and parameter design of the super-capacitor is very important. If the capacity setting is too small, the excess energy at the DC bus side during LVRT may not be absorbed, and the system requirement cannot be met. If the setting is too large, it may cause waste and increase economic costs. For a system with a rated power of 5 kW, the power transfer efficiency of the isolated bidirectional DC converter is about 75%. Assuming the capacitance of the super-capacitor is $C_{sc}$, the initial voltage is $U_{sc\_init}$, and the rated voltage is $U_{sc\_final}$, the charge and discharge power at the maximum instantaneous moment is expressed by Formula (8).

$$P_{sc\ max} \geq P_{r\ max} \cdot \eta = 5000W \times 0.75 = 3750\ W \tag{8}$$

The rated voltage on the DC side is 540 V, the super-capacitor voltage is set to 150 V, and the LVRT time is set to 0.4 s. According to Formula (9), the capacity of the super-capacitor can be deduced.

$$C_{sc} \geq \frac{2\eta \cdot P_{r\ max} \cdot T_{lvrt}}{U_{sc_{final}}^2 - U_{sc_{init}}^2} = \frac{2 \times 3750 \times 0.4}{540^2 - 150^2} F = 11.14\ mF \tag{9}$$

Consider a certain margin, set the rated capacity of the super-capacitor 12 mF. Firstly, to make the super-capacitor not overcharge or over-discharge in the process of LVRT, the energy storage system is protected. Secondly, considering the current withstand capability of the power electronic device in the converter, the operating voltage is set higher. It can protect these power electronics from over-current and breakdown damage.

Filter Inductor $L_{sc}$ Design

The design value of filter inductance needs to meet the working requirements of both the Buck and Boost circuits [24]. At the early stage of LVRT, a bidirectional DC/DC converter operates in Buck mode and is charged to the super-capacitor utilizing DC bus capacitors. In the recovery phase of LVRT, the bidirectional DC/DC converter operates in Boost mode, and the super-capacitor is used. The constant current discharge mode feeds back the stored energy to the DC bus $U_{dc}$. Since the average current in the loop mode is lower than the maximum current in the Buck mode, the filter inductor size can be designed according to the maximum ripple current in the loop mode of the Buck mode, as Formula (10)

$$L_{sc} = \frac{D \cdot (U_{dc}/n - U_{sc\_init})}{\Delta I_{scp} \cdot f_s} \tag{10}$$

$U_{dc}$ is the rated voltage of the DC bus. $U_{sc\_init}$ is the initial voltage of the super-capacitor (150 V). $n$ is the ratio of the isolation transformer. $D$ is the duty cycle in the Buck mode. $I_{scp}$ is the peak value of the rated current of the super-capacitor in the Buck mode. $\Delta I_{scp}$ is the maximum ripple current allowed by the circuit, usually set to 15% of the peak current rating. $f_s$ is the switching frequency of the tube in a bidirectional DC/DC converter.

*4.2. Fuzzy Adaptive Control Strategy during LVRT*

Fuzzy control theory is a control strategy for studying fuzzy phenomena and belongs to nonlinear control algorithms. The main structure of fuzzy control includes fuzzification, knowledgebase, fuzzy reasoning, and clarity. Figure 6 shows the DC bus voltage and current waveforms of the converter before and after the energy storage configuration of the LVRT fault process.

It can be seen from the waveform that the DC bus voltage is stable after the energy storage, at 540 V. During the period of 0.1–0.4 s, the analog grid voltage drops, and the battery is in discharge until grid voltage is restored at 0.4 s, thus the DC bus voltage is maintained stable. The battery absorbs the energy when the DC side voltage fluctuates greatly to ensure smooth progress during LVRT. The existence of the energy storage not only ensures LVRT ability but also reduces the voltage fluctuation at the DC bus terminal. The voltage fluctuates at 540 V, and the maximum does not exceed 541 V.

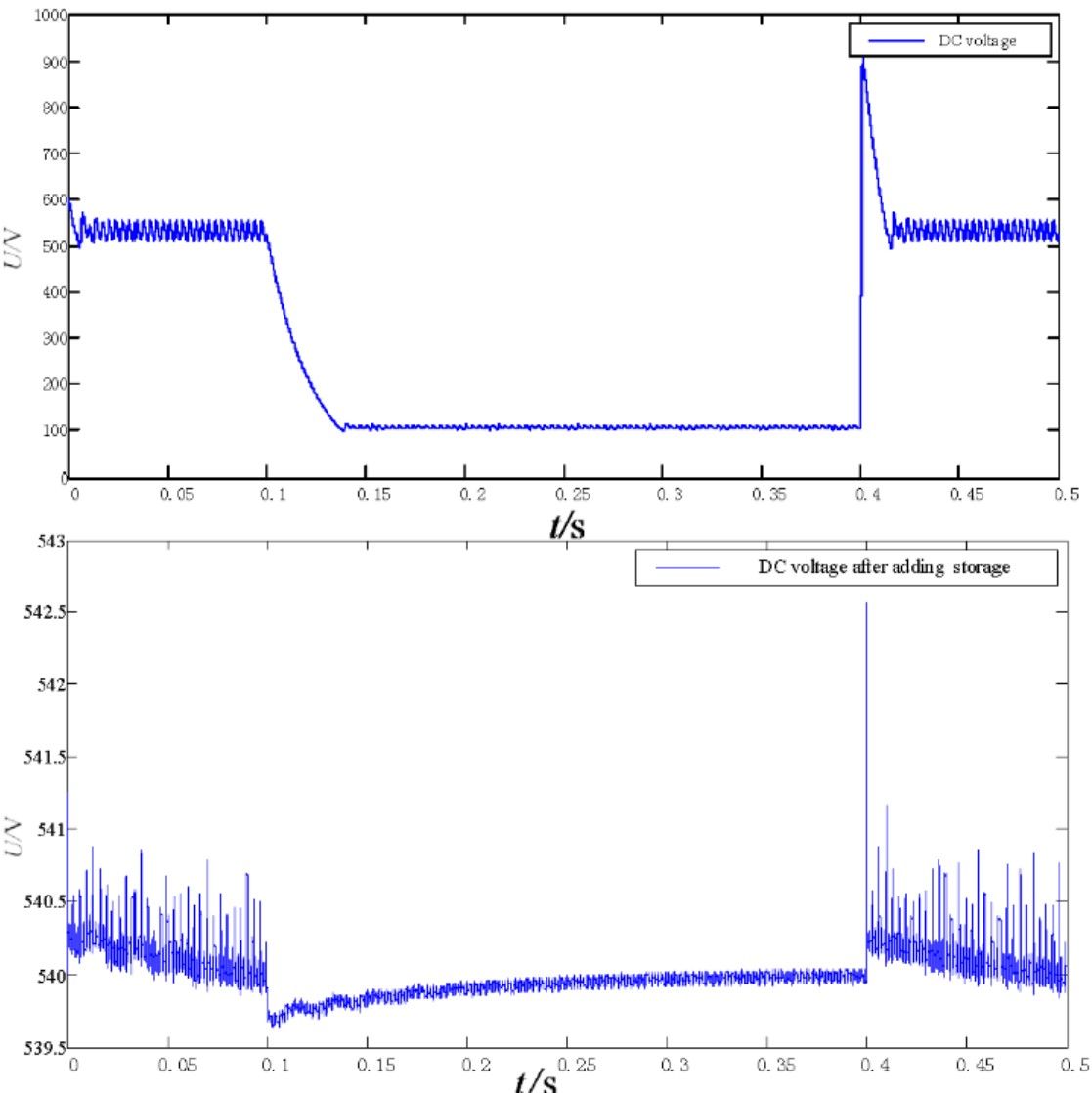

**Figure 6.** DC bus voltage during LVRT with storage.

*4.3. Reactive Compensation during LVRT*

According to power system analysis [25], the relationship curve between system voltage and reactive power can be expressed by Formula (11), where $U$ represents the system voltage, $Q$ represents the reactive power of the load, $U_o$ is the system voltage on the condition that $Q$ is 0, and $S_{sc}$ is the short-circuit capacity of the system.

$$U = U_0(1 - \frac{Q}{S_{sc}})$$

(11)

In the case that fluctuations of loads are small, the system voltage is greater than the load voltage fluctuates, and the line resistance is smaller than the impedance, the relationship of reactive power and voltage is approximately linear. When the electric load peaks periods arrive or LVRT occurs, so that the grid voltage drops while the reactive power is reduced, to support the reactive power required by power transmission, it is necessary to adjust the phase difference between the voltage and the current to change the power factor and provide reactive power compensation. Static Var Compensator (SVC) is a common dynamic reactive power compensation device, which adjusts output reactive power by changing its capacitive or inductive equivalent reactance, to achieve the purpose of controlling the corresponding bus voltage or reactive power of the system. When the system voltage is low or high,

by controlling the trigger time of the switch tube, changing the trigger angle, thereby changing the size of the access admittance, the purpose of changing the amount of reactive power can be achieved, so that voltage returns to the normal value. In this topic, the goal of reactive power compensation is achieved through the on–off control of the DC/AC converter at the interface between the energy storage and the grid, as shown in Figure 7.

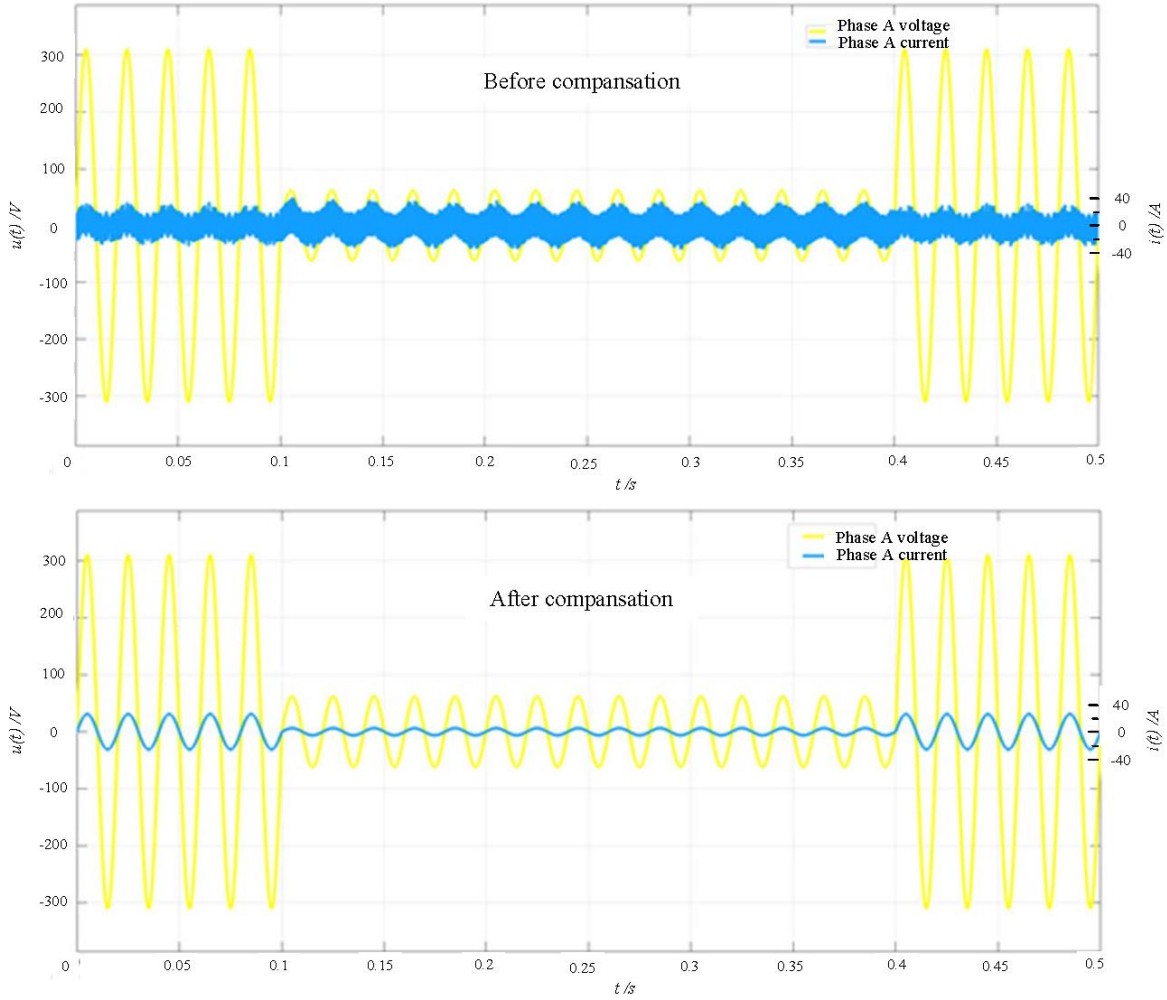

**Figure 7.** Reactive compensation during LVRT.

There is a phase deviation between the voltage and current without reactive power compensation, whether or not experienced by LVRT. After reactive power compensation, the voltage and current are in the same phase, and the system operated in unit power factor, realizing the aim of reactive power compensation during LVRT.

## 5. Active Management Energy Coordination Control Algorithm and System Simulation

Figure 8 is a group load operation active demand management and coordination control flow chart. The management scheme developed is focused on active energy management on the demand side. Based on historical data and real-time monitoring parameters, various converters among energy storage, grid, and load are reasonably controlled to realize load-friendly power supply. In the algorithm, considering the grid malfunction and the *SOC* of the energy storage and the peak load of the grid, the energy storage dispatch is used rationally to effectively avoid the peak power consumption.

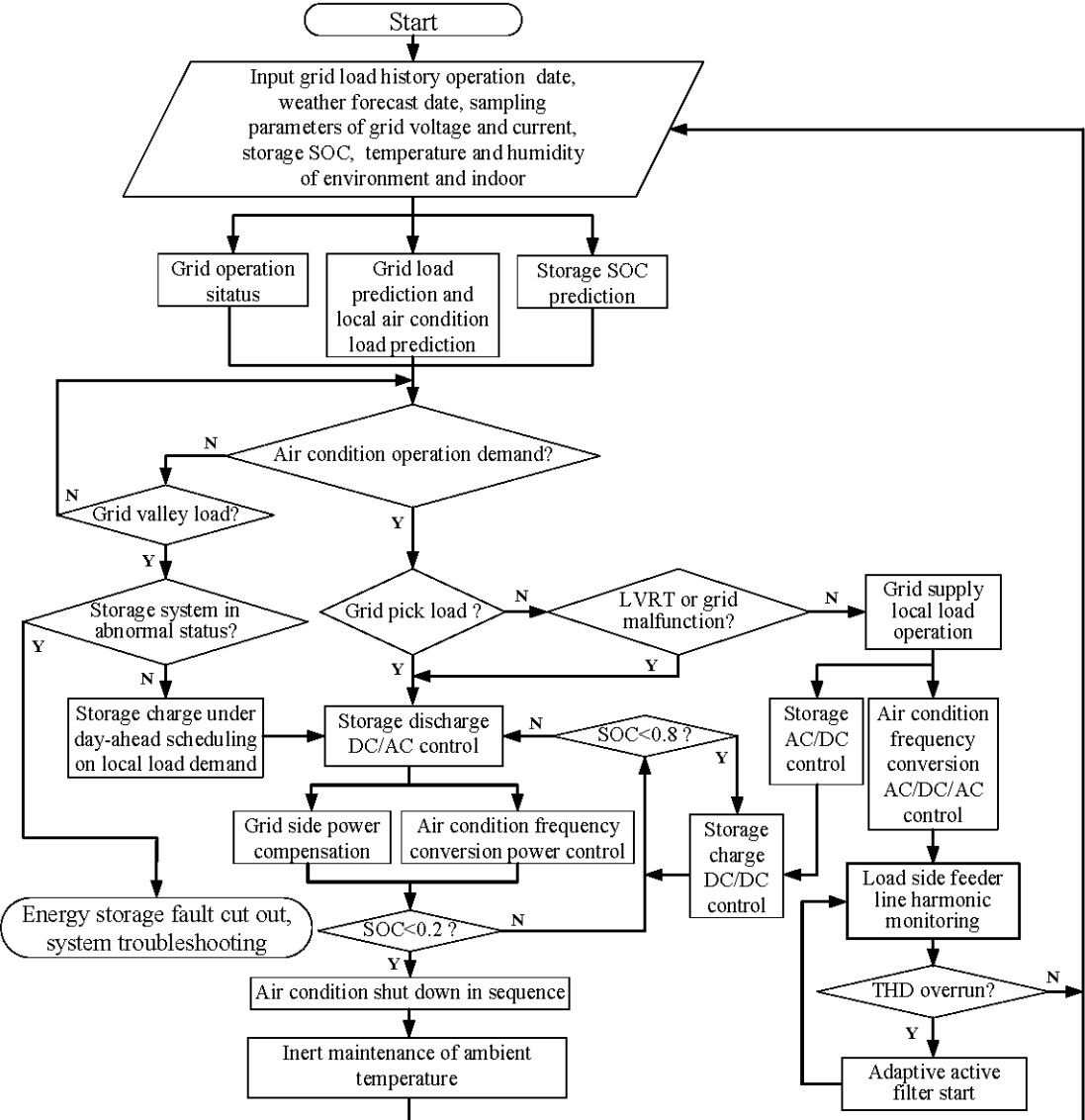

**Figure 8.** Active demand management of group load operation coordination control flow chart.

The technical points are as follows:

(1)　Historical data of grid load operation are employed to predict grid load change. Based on weather forecast temperature and humidity, the local air conditioning load operation data are predicted.

(2)　According to the monitored *SOC* data, the chargeable and dischargeable power of the energy storage system are predicted.

(3)　Combining grid load forecasting and local load forecasting data, day-ahead power scheduling is used to achieve storage energy management.

(4)　With real-time monitoring of the operating status of the air conditioner and the grid load, considering events of the grid load peak period and grid malfunction or LVRT, the flexible entry of energy storage system supply on the air conditioning operation is set up.

(5)　Once the grid is operating normally and experiencing a non-peak load period, the local air conditioning loads are supplied by the power grid. At the same time, if detected *SOC* is lower than 0.8, charging behavior is triggered to ensure that the function of the energy storage system can play normally.

(6) When the *SOC* is lower than 0.2, considering elapsed a long time of the temperature and humidity change, air conditioning loads can be stopped temporarily to pass the load peak period. Then, it is restarted to realize the staggering power supply.

(7) The storage energy supplement is carried out timely during off-peak periods to ensure providing reactive power compensation ability to the grid in the event of LVRT or grid malfunction.

(8) During the power grid supplying local air conditioning load, judged by monitoring the current distortion rate in the feeder line of the load circuit, the adaptive APF is put into use to eliminate higher proportion harmonics.

The simulation system corresponding to the coordinated control algorithm is shown in Figure 9. Table 3 shows the main simulation parameters. In the system, dynamic monitoring of grid parameters is done, and current and voltage are detected and analyzed.

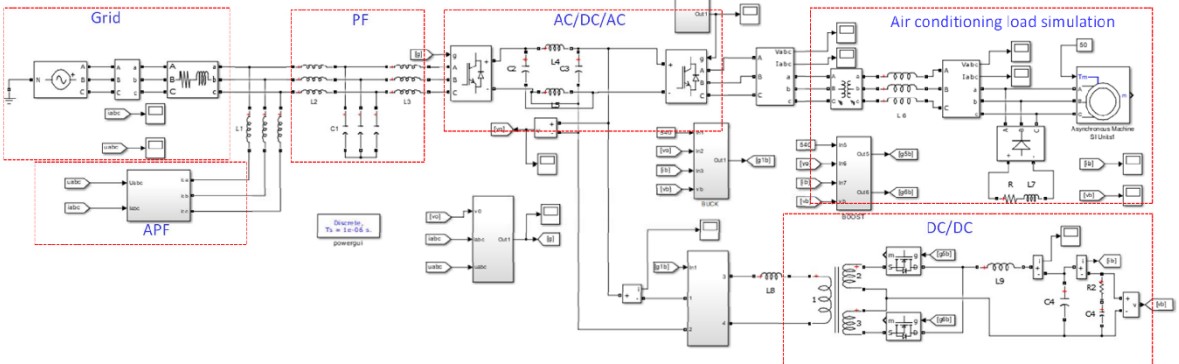

**Figure 9.** Active demand simulation circuit.

**Table 3.** The main simulation parameters.

| Components | Symbols | Value |
|---|---|---|
| Three-phase voltage | $U_a$, $U_b$, $U_c$ | 220 V |
| Motor frequency | f | 50 Hz |
| Motor 1 power | $P_1$ | 746 W |
| Motor 2 power | $P_2$ | 3000 W |
| Motor 3 power | $P_3$ | 4500 W |
| Rectifier side resistor | R | 10 Ω |
| Rectifier side inductor | $L_7$ | 0.002 H |
| APF hysteresis width | w | 0.001 A |
| PF cutoff frequency | $f_L$ | 2.25 kHz |
| Filter capacitor | $C_1$ | 0.282 mF |
| Filter inductor | $L_2$, $L_3$ | 17.76 uH |
| Storage Super-capacitor | $C_4$ | 12 mF |
| Storage filter inductor | $L_9$ | 10 mH |

The simulation system used for active demand-side management research is composed of a power grid, PF, APF, AC/DC/AC converter, DC/AC converter, and load. The filters are connected in parallel with the front end of the frequency converter, to actively prevent the grid from being damaged and influence the harmonics caused by the load operation. The energy storage system is connected to the DC bus line of the AC/DC/AC converter through the DC/DC converter. According to the grid monitor status, AC/DC/AC converter working mode is controlled. Once LVRT or grid malfunction happens, the energy of the storage system can flow to the grid and the load side at the same time through the coordination control. Among them, the DC/AC supplies load, the bi-directional AC/DC works as an inverter, providing reactive power compensation to the grid.

## 6. Results and Discussion

Figure 10 is the voltage waveform monitored from the DC bus of the energy storage interface and current waveform of the grid side. The left is voltage waveforms that happened at the time of nonlinear group load normal operation, LVRT beginning, LVRT process, and LVRT ends, respectively. The right is the three-phase power supply current waveform connected to the load that goes through this process. The voltage amplitude fluctuates at 540 V, and the maximum value does not exceed 541 V. The current amplitude is 5 A, and the phase of each current is two-thirds π radian apart. With reactive compensation, the system passes LVRT smoothly

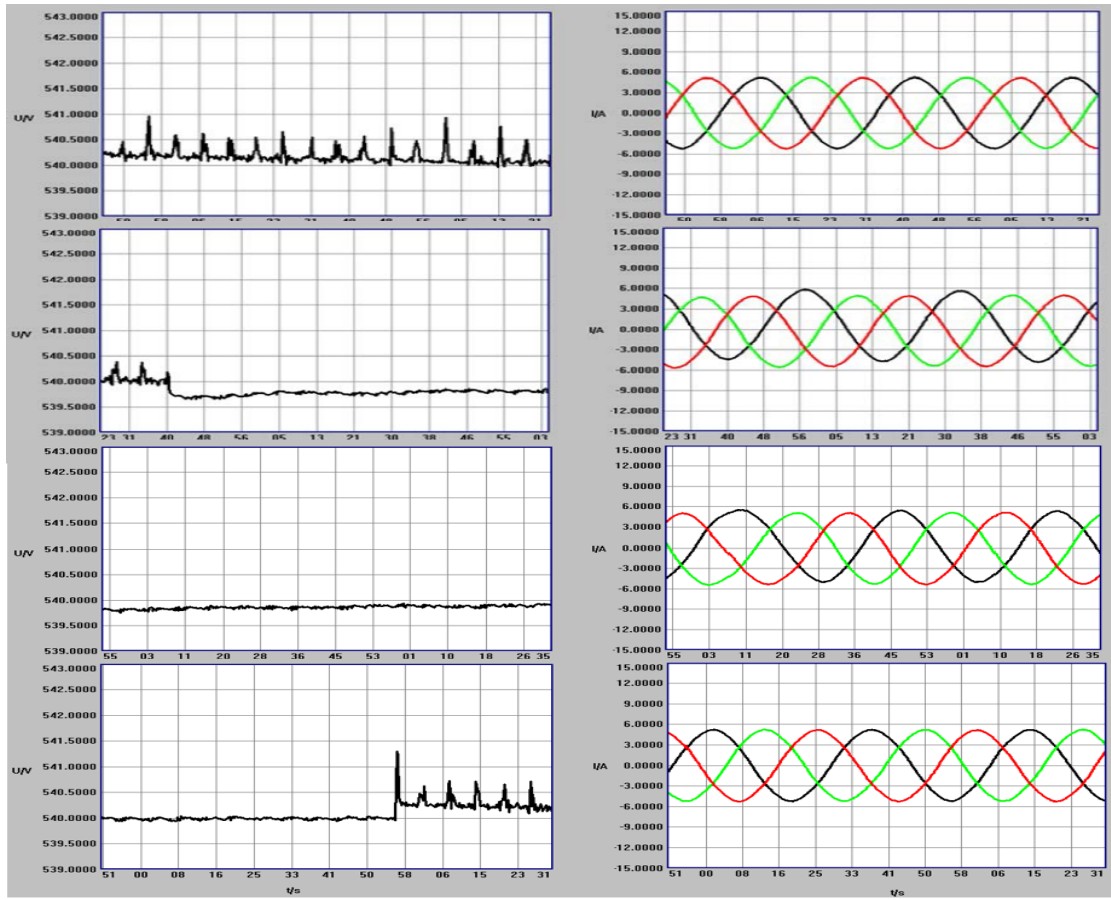

**Figure 10.** Waveforms of DC bus interface voltage and current on the grid side.

Limited by the speed of serial communication, to solve the problem of complete display on the upper computer of the data collected by the lower computer, some of the data collected by the lower computer are firstly stored in the Excel table, and then the upper computer reads them from Excel table and then displays them; thus, the abscissa in the figure is the change value of the second hand of the actual clock, which is displayed cyclically from 0 to 59 s. The abscissa in the left figure shows the time of the LVRT process, but the figure on the right does not reflect the true sinusoidal frequency. The true sinusoidal frequency is 50 Hz. The current date on the right is from the Excel table, where the real sampling period from the lower computer is 10us, and the collected complete sine cycle contains 5000 data. However, the read data period of the host computer from the Excel table is 1.75 ms, and the sine waveform is continuously refreshed.

The figure also shows that, when the grid is normally supplied local load, due to grid voltage fluctuation, there are continuous small fluctuations in the DC bus line of AC/DC/AC. When an LVRT occurs, the air conditioning loads are directly supplied by the energy storage inverter system, which is connected to the DC bus; obviously, the voltage fluctuation of the DC bus during the LVRT process is

smaller than before. Furthermore, the DC bus voltage drops no more than about 0.35 V. Once the LVRT process ends after 311 s, the original volatility state is reproduced.

On the other hand, it can also be seen from the right current waves that, whether before or after LVRT, few harmonic components are produced. The simulation results also show that, during the period, the maximum current amplitude increases to 6 A, which is represented by the black and red lines, and then gradually recovers. Thus, it is verified that the filter design can actively suppress the harmonics caused by the nonlinear load to the grid. With energy storage participating in grid fault control, DC bus voltage can be maintained constant during the grid LVRT. Once the grid is restored, the load resumes normal grid power supply. Waveforms also show that harmonic suppression and LVRT prevention designed can play a good role in the grid fault. An active demand management function can also be implemented.

In this manuscript, a new active demand-side management strategy is proposed for the air conditioning group load operation. A collaborative control scheme is given. The research focuses on harmonic active suppression and LVRT technology, reactive compensation, and simulation software is used to design the system. The summary is as follows.

1.  By studying the characteristics of the group air conditioning operation, a hybrid model of the electronic circuit is determined with similar nonlinear characteristics as the load model.
2.  By studying the influence characteristics of the air conditioning load model operation on the power grid, for the harmonic characteristics, the hybrid application of the double hysteresis APF and the PF are introduced to weaken the influence of harmonics on the grid actively.
3.  Considering the influence of grid LVRT on the VFC load, an active emergency prevention mechanism with a dynamic energy storage switch is proposed to ensure the stable operation of the load.
4.  According to the complex composition of the system, the sensor monitoring network strategy is proposed, and the upper dynamic monitoring design is implemented to realize the system coordination operation, reducing the impact of load operation on the grid.

## 7. Conclusions

This article totally explores an active demand energy management strategy. Compared with other active demand management strategies that focus on economic benefits, this article pays more attention to the management of power quality at the end of power consumption. Based on the analysis of the complex characteristics of group load operation accessed to the grid, comprehensive consideration is given under such cases as outburst faults and quality factors caused from the grid side and load side, where the special working characteristics of the energy storage system are fully used, and the peak–valley shifting requirement of the power grid is also taken into account. Correspondingly, the designed control strategy collaboratively focuses on filtering technology and energy storage technology, and the feasibility is verified by simulation. The results have some application and promotion value. Of course, some work needs to be further studied, for example energy management of storage systems for dual hysteresis power supply and LVRT active prevention, depth study of energy-saving technology for VFC operation, and application of the hybrid intelligent algorithm in harmonic processing. These will be studied in the future.

## 8. Patents

Chinese invention patents are resulting from the work as follows:

1.  "Active demand strategy based on power-friendly air conditioning load side". Authorization number: ZL201510243894.1. Authorization date: 2017/05/10.
2.  "Effective energy-saving active demand method based on power-friendly air-conditioning load side". Authorization number: ZL201710100461.X. Authorization date: 2019/06/18.

3. "Active demand method based on power-friendly air-conditioning load side to reduce system cost". Authorization number: ZL201710239792.1. Authorization date: 2019/01/29.
4. "Design method of active demand energy storage for nonlinear group air-conditioning group load operation". Application Number: 2018082401656550. Now in the patent actual review stage.

**Author Contributions:** The article is finished by three authors, their contributions are as follows: J.-h.Z., methodology formal, analysis and writing-original draft preparation; J.G., investigation, project administration and funding acquisition, writing-review, and editing; M.W., software validation. All authors have read and agreed to the published version of the manuscript.

**Funding:** This research was funded by the National Natural Science Foundation of China [Grant No.61673226] and the Jiangsu Provincial Department Natural Foundation of China [Grant No.15KJB470014, Grant No.18KJA470003].

**Acknowledgments:** This research was funded by the National Natural Science Foundation of China [Grant No.61673226] and the Jiangsu Provincial Department Natural Foundation of China [Grant No.15KJB470014, Grant No.18KJA470003]. Nantong Science and Technology Bureau Project of China [Grant No. JC2018116]. The authors would like to thank the supports from both the Ministry of Science and Technology and the National Natural Science Foundation of China.

**Conflicts of Interest:** The authors declare no conflict of interest.

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
