# Peer review of "Grid-Friendly Active Demand Strategy on Air Conditioning Class Load"

_applsci, doi:10.3390/app10186464_

Round 1

Reviewer 1 Report

This paper deals with a new technology to reduce harmonics caused by air conditioning systems.

The literature review in the introduction should be extended.
More recent papers (from 2018 onwards) should be cited.

Equation (1) in section 2 raises a lot of questions.
First of all, the ssd (human body comfort index) does not seem to be a standard index.
I was unable to find it in reference [10] as stated.
Maybe, it can be found in [9]. However, that paper has been retracted. Therefore, that paper shouldn't be cited in any other paper anymore.
In equation (1), values with units and dimensionless values are mixed.
The relation between ssd and Wi is not clear either. Equation (1) therefore looks like two different equations.
On the other hand, equation (1) does not seem to be required for the rest of the paper anyway, so you might just remove it.

Fig. 1 is not clear. What do the arrows mean? Can it be simplified?

Table 1: Why does the last row (THD) only contain one entry?
The values given in lines 118 to 122, could be included in Table 1 for a better comparison of limits and actual values.

The quality of figures 1, 2, 4 and 7 should be improved.

Reference [8] does not seem to exist. Is there a URL to it?

A thorough language check is recommended. For instance, what does "As a thumb" mean in line 57?

Author Response

Dear Editor and Reviewer

Many thanks for your email of 21 August 2020. Thank you for your valuable comments. After carefully studying the comments and your advice, we have made corresponding changes to the paper. The manuscript is revised submission (Applied Sciences, Manuscript ID: applsci-880230, title Grid-friendly Active Demand Strategy on Air Conditioning Class Load. Furthermore, the relevant regulations had been made in the original manuscript according to the comments of reviewers, and the major revised portions were marked in red. We also responded point by point to your kind advices and reviewer’s detailed suggestions as listed in attach.

Besides the above changes, we have corrected some expression errors.

We sincerely hope this manuscript will be finally acceptable to be published on “Applied Sciences”. If you have any question about this paper, please don’t hesitate to let me know.
Thank you for your consideration.
We are Looking forward to hearing from you soon.
With kindest regards!

Yours Sincerely

Jian-hong Zhu

Reviewer 2 Report

Figure 1: it is tnot clear where the dynamic load monitoring tets his information.

Figure 9 needs more expalnation

A comparison between the effekt of the different methods/alternatives and the effort therefore is missing

Author Response

(The authors gave the same response as above.)

Reviewer 3 Report

The paper is devoted to a very interesting topic related to the active management of air conditioning loads.

However, the paper presents several issues that have to be improved:

  1. English language and style should be checked;
  2. The punctuation should be revised (mostly in the abstract where the semicolon makes very difficult the reading);
  3. State of the art is very poor:
    1. the authors do not describe sufficiently the context: demand-side management strategies and their use are not presented; the impact of air conditioning loads in the daily load demand is not described;
    2. even if power-quality is the focus of the study, a sufficient number of references is not provided.
  4. Figures and flowchart are not well described:
    1. figure 4 is the second figure cited (thus it should be figure 2),
    2. the flowchart in figure 7 is not explained,
    3. Figure 8 is too small and does not add useful information to the reader. Is Figure 8 a portion of a network? Is it the control scheme?
  5. In Chapter 5 (Active management energy coordination control algorithm and system simulation), the active management algorithm is not described. Only a flowchart is proposed. Is it a local/decentralised/centralised approach? Has it ever been applied? Is it just a proposal? Please explain.
  6. The case study is not adequately presented and described; it is just summarised through a table without providing any useful information
  7. Some sentences should be better explained:
    • "...the lack of reactive power at peak load can cause voltage fall and even lead to entire system collapse, which undoubtedly has a great impact on the normal daily life of the people." 
    • "Based on the approximate circuit model design of the nonlinear group load of air conditioning, the energy characteristics of the load under the ideal grid condition are monitored."

Author Response

(The authors gave the same response as above.)

Round 2

Reviewer 1 Report

Some of my comments have been addressed.
However, the issues around equation (1) remain.
Please clarify or make the necessary changes.

Author Response

Dear Editor and Reviewer

Many thanks for your valuable comments in the email of 3 September 2020. After carefully studying the comments and your advice, we have made corresponding changes to the paper. The manuscript is revised submission (Applied Sciences, Manuscript ID: applsci-880230, title Grid-friendly Active Demand Strategy on Air Conditioning Class Load. Furthermore, the relevant regulations had been made in the original manuscript according to the comments of reviewers, and the major revised portions were marked in red. We also responded point by point to your kind advices and reviewer’s detailed suggestions as listed below.

Besides the above changes, we have corrected some expression errors.

We sincerely hope this manuscript will be finally acceptable to be published on “Applied Sciences”. If you have any question about this paper, please don’t hesitate to let me know.
Thank you for your consideration.
We are Looking forward to hearing from you soon.
With kindest regards!

Yours Sincerely

Jian-hong Zhu

Reviewer 3 Report

Dear Authors,

thank you for your effort for improving the paper. However, some work still need to be done.

1) English language has to be checked throughout the text. There are some error and typos that need to be fixed like in line 99, in Figure 7 compAnsation.

2) Table 3, please indicate the unit of measure according with the universal standard (e.g., Three-phase voltage/V --> Three-phase voltage [V
] or Three-phase voltage 220 V); moreover check the capitalisation and the space between the number and the unit (for instance in line 400: 541v --> 541 V).

I deeply suggest that the Authors get editing help from someone with full professional proficiency in English.

3) The authors introduce the paper talking about "active demand-side management", but no definition is provided (some paper are cited but not for explaining the concept).

4) Figure 9 is still too small and does not add further information to the reader.

5) A paragraph in which the outline of the paper is described could be helpful for the reader in order to understand where the topics will be described (there is something similar in the abstract).

6) There is not a Conclusion section (conclusions are included in 6. Results and Discussion).

Author Response

(The authors gave the same response as above.)

Round 3

Reviewer 1 Report

Thank you for addressing my comments.
I suggest to add units to the summands in equation (1) to make it clear what is being summed up.

Reviewer 3 Report

The Authors made the improvements suggested.